# Comparative Efficacy of Velopharyngeal Surgery Techniques for Obstructive Sleep Apnea: A Systematic Review

**DOI:** 10.3390/medicina59061147

**Published:** 2023-06-14

**Authors:** Ana Maria Vlad, Cristian Dragos Stefanescu, Iemima Stefan, Viorel Zainea, Razvan Hainarosie

**Affiliations:** 1“Prof. Dr. Dorin Hociota” Institute of Phonoaudiology and Functional ENT Surgery, 21st Mihail Cioranu Street, 061344 Bucharest, Romania; 2ENT Department, Faculty of Medicine, “Carol Davila” University of Medicine and Pharmacy, 030167 Bucharest, Romania; 3Medical Center of Special Telecommunications Service, 060044 Bucharest, Romania

**Keywords:** obstructive sleep apnea, palatopharyngeal surgery, barbed reposition pharyngoplasty, expansion sphincter pharyngoplasty, uvulopalatopharyngoplasty

## Abstract

*Background***:** In recent years, surgical interventions for obstructive sleep apnea (OSA) have evolved rapidly, with numerous techniques described in the literature. The approach to velopharyngeal surgery for obstructive sleep apnea has transformed over time, shifting from an aggressive removal of redundant excess soft tissue to less invasive reconstruction techniques that aim to preserve pharyngeal function while effectively managing sleep apnea. This review aims to evaluate and compare the efficacy of the surgical techniques utilized for OSA at the level of the palate and pharynx. It will cover both traditional and novel procedures. *Methods:* A comprehensive search of the major databases, such as PubMed/MEDLINE, Web of Science, and Scopus, was conducted to identify the relevant literature. We included articles written in English that analyzed the outcomes of adult patients who received velopharyngeal surgery for sleep apnea. Only comparative studies that examined at least two techniques were considered. *Results:* In all of the studies combined, the total number of patients who underwent velopharyngeal surgery was 614 in eight studies. All surgical procedures resulted in improvements in the apnea–hypopnea index (AHI). The highest success rates and best outcomes were achieved by barbed reposition pharyngoplasty (BRP) in most studies, ranging from 64.29% to 86.6%. BRP also demonstrated the most significant improvements in both objective and subjective parameters closely followed by ESP that obtained similar efficiency in some studies, especially when combined with anterior palatoplasty (AP), but with a higher incidence of complications. While LP showed moderate efficiency compared with BRP or ESP, the UPPP techniques exhibited greater outcome variability among studies, with a success rate ranging from 38.71% to 59.26%, and the best results observed in a multilevel context. *Conclusions:* In our review, BRP was the most preferred, effective, and safe among all velopharyngeal techniques, closely followed by ESP. However, older described techniques also showed good results in well-selected patients. Larger-scale studies, preferably prospective, that rigorously incorporate DISE-based strict inclusion criteria might be needed to assess the efficacy of different techniques and generalize the findings.

## 1. Introduction

It is estimated that worldwide, 936 million adults suffer from sleep apnea with important social and economic burdens secondary to the major complications it has on health [1]. 

Obstructive sleep apnea is characterized by recurrent upper airway collapse during sleep, leading to apnea/hypopnea with oxygen desaturation episodes. Various methods have been proposed for assessing upper airway (UA) obstruction. Drug-induced sleep endoscopy (DISE) is now widely regarded as the most effective approach for accurate localization of the collapse areas requiring targeted treatment planning [2]. Usually, multiple sites of obstruction are observed during DISE, with the most observed type of collapse at the palatal level [3]. CPAP remains the preferred initial treatment option with the strongest evidence of efficacy, as it can effectively open the upper airway at all levels. However, its limited compliance and acceptance rates make it necessary to explore alternative therapies [4].

For more than 40 years, since its first introduction in 1981 by Fujita et al. [5], UPPP remained one of the most common procedures performed for upper airway collapse, typically utilized as part of a multilevel approach due to inconsistent outcomes seen in single-level surgery [6]. The evaluation of a patient’s suitability through several preoperative assessments to determine good patient selection might result in a more successful UPPP. However, despite careful patient selection and favorable outcomes in some cases, the procedure still carries a high risk of complications [7]. Anterior palatoplasty (AP) [8] and uvulopalatal flap (UPF) are other, similar procedures that address retropalatal obstruction, with favorable results in selected patients usually in mild to moderate sleep apnea [9].

With an increased understanding of palatopharyngeal anatomy, a move away from non-selective, resective procedures toward more refined and individualized treatment approaches has been adopted. Cahali et al. in 2004 were the first to show that addressing the lateral pharyngeal walls is necessary to achieve more positive surgical results [10]. Through his procedure, superior pharyngeal constrictor muscle is microdissected within the tonsillar fossa and cut, resulting in a laterally based flap of muscle that is attached to the palatoglossus muscle on the same side. However, dysphagia was an important issue, and a new technique was proposed a few years later. The expansion sphincter pharyngoplasty (ESP) technique isolates and rotates the palatopharyngeal muscle while leaving the superior pharyngeal constrictor muscle intact [11]. This determines the pulling of the muscle in a superoanterolateral direction with a less invasive approach. This proved to be an effective procedure, especially in patients with lateral wall collapse, determining fewer complications. In recent years, modified pharyngoplasties utilizing barbed sutures, also referred to as barbed pharyngoplasties (BPs), have been developed. The technique described by Vicini et al. in 2015 (barbed reposition pharyngoplasty) [12], involves repositioning the posterior pillar, specifically the palatopharyngeal muscle, to a more lateral and anterior location in order to increase the size of both the oropharyngeal inlet and the retropalatal space. Several studies reported excellent results with this technique, with minimal complications. 

There has been a significant increase in the development of various procedures and modifications of surgical techniques to achieve the best possible results tailored to the individual characteristics of the upper airway. This systematic review aims to gather and explore all available evidence on the effectiveness and safety of different surgical techniques for treating obstructive sleep apnea in adults and provide insights into which technique may be the most effective and safe for patients. The analysis takes into account both objective parameters such as AHI and subjective parameters such as ESS. Success rates are reported for each technique, and trends in their usage are discussed. 

## 2. Materials and Methods

### 2.1. General Study Design

This systematic review was carried out in compliance with The Preferred Items for Systematic Reviews and Meta-Analyses (PRISMA) guidelines [13].

### 2.2. Selection Criteria

The review was conducted using the PICOs protocol and encompassed studies that compared various techniques utilized in palatopharyngeal surgery, as follows:

(P): Population: Adult patients diagnosed with obstructive sleep apnea (OSA), undergoing palatopharyngeal surgery

(I): Intervention: Comparison of two or more surgical techniques used in palatopharyngeal surgery for OSA patients, such as, but not limited to, uvulopalatopharyngoplasty (UPPP), barbed reposition pharyngoplasty (BRP), expansion sphincter pharyngoplasty (ESP), lateral pharyngoplasty (LP), or variations of these techniques.

(C): Comparison: Pre- and post-treatment outcomes of the different surgical techniques used.

(O): Outcome:

Primary outcomes: Assessment of treatment efficacy, including improvements in apnea–hypopnea index (AHI) and success rate, 

Secondary outcomes Epworth Sleepiness Scale (ESS)*,* complications.

(s): Study design: Both prospective and retrospective studies.

The exclusion criteria for the study were defined as follows:Studies on the pediatric population.Studies not in English.Reviews, meta-analyses, editorial letters, technical notes.Studies with insufficient or missing data.Studies that did not analyze AHI.Studies that did not compare at least two different palatopharyngeal surgical techniques or that compared variations of the same technique.Studies that presented outcome variables (such as AHI) as an average rather than for each individual technique.

### 2.3. Search Strategy

Systematic electronic searches were performed by two different authors (A.M.V. and I.S) on PubMed/MEDLINE, Web of Science, and Scopus databases to identify relevant studies. The search strategies included different combinations of the following descriptors and/or medical subject headings (MeSH): (“palate surgery” OR “soft palate surgery” OR “uvula surgery” OR “Uvulopalatopharyngoplasty” OR UPPP) AND/OR “uvulopalatal flap” AND/OR (“lateral pharyngoplasty” OR “Cahali lateral pharyngoplasty”) AND/OR (“Expansion sphincter pharyngoplasty” OR ESP) AND/OR (“Barbed reposition pharyngoplasty “OR barbed suture* OR BRP) AND (“sleep apnea” OR “obstructive sleep apnea” OR “OSA” OR “OSA surgery”). The search terms were adapted to the particular requirements of each database. Only studies that had been published within the last 5 years were considered for inclusion. The last search was conducted on 30 March 2023.

The search results obtained from each database were imported into the reference manager software Endnote, to manage and organize the articles. Duplicates were identified and subsequently removed.

### 2.4. Data Extraction

Initially, all articles underwent screening based on their titles and abstracts. Subsequently, the full-text versions of each publication were evaluated, and those considered unrelated to the scope of this review were excluded. The selected studies underwent independent evaluation by two investigators (A.M.V and I.S.), and necessary data were collected. The information extracted from the selected studies included: the name of the authors and the year of publication, study design, sample size, patients’ profiles, surgical techniques compared, mean follow-up period, and objective or subjective outcomes (AHI, ESS). Any disagreement between the authors was discussed and resolved through consensus after consultation with the senior reviewer (C.D.S).

### 2.5. Statistical Analysis

Statistical analysis was conducted using the Jamovi software 2.3.26. The analysis was carried out using the mean difference as outcome measure to compare pre- and postoperative apnea–hypopnea index (AHI) and Epworth Sleepiness Scale (ESS) outcomes. We adopted a random effects model to estimate effects measures by 95% confidence interval. Forest plots for each outcome were provided. A total of 7 studies were included in the AHI analysis, and 6 studies were included in the ESS analysis. The Q-test and the I^2^ statistic were calculated to assess the presence of data heterogeneity between studies. To compare the surgical techniques (UPP, ESP, and BRP), subgroup comparisons were performed. Binary variables were created to represent the utilization or non-utilization of each technique in the studies compared. For example, in the BRP vs. UPP comparison, a binary variable was assigned a value of 1 for sub-studies that performed the BRP technique and 0 for UPP studies. Similar binary variables were constructed for the other comparisons. These binary variables served as moderators in the analysis, enabling a direct comparison between the techniques while considering the specific technique employed in each study.

## 3. Results

### 3.1. Study Selection

The authors identified a total of 143 potentially suitable studies through the search strategy presented in the Methodology section. The studies’ selection steps are summarized in Figure 1. After eliminating the duplicates through the Endnote reference manager, a total of 70 articles were analyzed regarding the title and abstract, applying the selection criteria to find the most appropriate studies for the review. All reviews, meta-analyses, editorial letters, and all studies that did not compare at least two surgical techniques were excluded, resulting in 23 full-text papers examined for eligibility. After removing the studies with incomplete or inappropriate information, eight studies were considered qualified for data extraction. 

### 3.2. Study Characteristics and Patient Profiles

Table 1 provides a synthesis of the main features of the studies included (Table 1). The majority of included studies were retrospective [14,15,16,17,18,19], while two of them were prospective [20,21]. The sample size of each study exhibited variability, with the number of subjects ranging from 20 [14] to 122 [18]. The total count of patients who underwent velopharyngeal surgery across all studies was 614, for which uvulopalatopharyngoplasty (UPP) and variations in the technique (m-UPP, RF-UPP) were described and performed in four studies (209 patients), anterior palatoplasty (AP) in two studies (52 patients), uvulopalatoflap placement in one study (23 patients), and lateral pharyngoplasty (LP/CLP) in two studies (41 patients). BRP and ESP were performed in five (166 patients) and four studies, respectively.

### 3.3. Outcomes

Table 2 provides a summary of the outcomes from all the studies that were reviewed.

#### 3.3.1. AHI Outcomes

All studies provided pre- and post-operative data regarding AHI. However, just seven studies were taken into account for meta-analysis since one of the studies expressed the results as median and quartile intervals instead of mean and standard deviation. The analysis was carried out using the standardized mean difference of AHI as the outcome measure.

A comprehensive database was created by subdividing each study based on the techniques and the corresponding outcomes. The random-effects model was fitted to the data, incorporating 18 sub-studies in the analysis. The observed standardized mean differences ranged from 0.1981 to 2.3633, with most estimates being positive (100%). The estimated average standardized mean difference based on the random-effects model was 1.0416 (95% CI: 0.7825 to 1.3007). Therefore, the average outcome differed significantly from zero (z = 7.8794, *p* < 0.0001), indicating a meaningful effect of the velopharyngeal techniques on the AHI.

According to the Q-test, the true outcomes appeared to be heterogeneous (Q(17) = 61.9697, *p* < 0.0001, tau^2^ = 0.2152, I^2^ = 71.5731%). However, despite this heterogeneity, the studies generally supported the estimated average outcome. An examination of the studentized residuals revealed that none of the studies had a value larger than ±2.9913, and hence, there was no indication of outliers in the context of this model. Additionally, none of the studies could be considered to be overly influential.

A forest plot was generated to visually display the mean differences between the velopharyngeal techniques in the included studies. The forest plot provides a clear overview of the effect sizes and their confidence intervals for each technique, allowing for easy comparison and interpretation (Figure 2).

#### 3.3.2. Subgroup Analysis

In the subgroup analysis, a random-effects model was applied to the UPP, BRP, and ESP techniques (Figure 3). These techniques were selected because they had sufficient studies available for analysis. However, it was not possible to perform a subgroup analysis for the other techniques due to limited data availability. Only one or two studies were available for those techniques, which did not provide enough data to conduct a meaningful comparison. Nevertheless, the results and trends of the other techniques were described and discussed based on the available studies.

##### Uvulopalatoplasty

The uvulopalatopharyngoplasty (UPPP) procedure was examined in four studies, and all studies showed a statistically significant reduction in the apnea–hypopnea index (AHI). In two studies [15,16,17], the performed technique was the classical one, described by Fujita et al. [5], and in the other two [16,17,18], the modified version according to Fairbanks et al. was used [22]. The analysis using the random-effects model for the UPP approach showed a mean difference of 1.24 (95% CI: 0.1694 to 1.0714) with an overall effect Z score = 2.6962 (*p* = 0.0070). Based on the Q-test results (Q(3) = 9.3795, *p* = 0.0246), it appeared that the outcomes observed in the studies were not consistently similar. The I^2^ value of 68.67% indicated a moderate level of heterogeneity.

##### Barbed Reposition Pharyngoplasty

In all six studies in which it was evaluated, the BRP technique obtained a statistically significant improvement in AHI. The analysis of five studies suggested that the BRP procedure led to a significant improvement in AHI based on the average difference observed across the included studies (z-value = 7.4986, *p* < 0.0001). The 95% confidence interval (CI) for this estimate ranged from 1.0466 to 1.7874, suggesting a high level of confidence in the results The analysis did not find significant variation in the true outcomes, as indicated by the Q-test (Q(4) = 6.8923, *p* = 0.1417, I^2^ = 43.4237%). This suggests that, overall, the studies showed similar results in terms of the effect of BRP on AHI improvement.

##### Expansion Sphincter Pharyngoplasty

ESP was addressed in four studies, with a statistically significant improvement of AHI in all of them. The estimated average standardized mean difference based on the random-effects model was 1.3308 (95% CI: 0.9272 to 1.7344), indicating a significant difference from zero (z = 6.4625, *p* < 0.0001). Regarding heterogeneity, the Q-test showed no significant amount of heterogeneity in the true outcomes (Q(3) = 5.9937, *p* = 0.1119, I^2^ = 40.3042%). This suggests that the true outcomes across the studies were generally consistent with the estimated average outcome, although some heterogeneity may have existed.

##### Comparisons between UPP, ESP, and BRP

Using moderator variables allowed us to assess whether the effect sizes (mean differences) varied depending on the technique used. When comparing the UPP and BRP techniques, the mixed-effects model analysis showed a significant difference in the mean AHI reduction (coefficient = 0.802, se = 0.295, Z = 2.723, *p* = 0.006, 95% CI: 0.225 to 1.380). Specifically, the BRP technique, used as moderator, was associated with a larger mean difference in AHI compared to the UPP technique. Similar results were observed for the ESP technique, which was associated with a larger mean difference in the outcome compared to the UPP technique (coefficient = 0.726, se = 0.324, Z = 2.241, *p* = 0.025, 95% CI: 0.091 to 1.362).

The positive coefficients suggested that the BRP and ESP techniques may be more effective in achieving the desired outcome compared to UPP. However, the use of BRP as a moderator did not significantly affect the AHI outcome when compared to ESP (coefficient = 0.216, se = 0.288, Z = 0.752, *p* = 0.452, 95% CI: −0.347 to 0.780). This suggests that there was no significant difference between the BRP and ESP techniques in terms of their impact on reducing AHI using the data from the studies.

##### The Other Techniques

Two studies evaluated lateral pharyngoplasty (LP) in comparison to other palatopharyngeal techniques [17,20]. Both studies reported a significant reduction in AHI (17.84 ± 13 from 46.3 ± 34.02 and 12.05 ± 15.23 from 17.69 ± 12.47). Anterior palatoplasty (AP) was described in two studies. The decrease in AHI value was statistically significant only in the Haytagiu et al. study (8.1 ± 7.3a from 17.5 ± 8.2) compared to in the Karakok et al. study (14.27 ± 15.43 from 16.90 ± 10.26). The UFP technique, mentioned in one study, was responsible for a great reduction in AHI (18.5 ± 7.9 to 8.6 ± 6.9).

#### 3.3.3. ESS Outcomes

The subjective outcomes for the techniques were measured in seven of eight studies using the Epworth Sleepiness Scale (ESS). Six studies had the necessary data to analyze the results. The estimated average standardized mean difference based on the random-effects model was 1.2631, with a 95% CI= 0.8877–1.6385 and a z-value= 6.5943 (*p*-value < 0.0001). This implies that surgery had a significant positive effect on the ESS subjective parameters. According to the Q-test, the true outcomes appeared to be heterogeneous (Q(14) = 83.3390, *p* < 0.0001, tau^2^ = 0.4385, I^2^ = 82.2159%). Hence, although the average outcome was estimated to be positive, in some studies the true outcome may in fact have been negative. A forest plot was generated to visually display the mean differences in ESS between the velopharyngeal techniques in the included studies (Figure 4).

#### 3.3.4. Success Rate

The success rate was determined in seven studies. The majority of studies used the Sher criteria of success rate (AHI reduction > 50 and AHI value < 20). For the BRP technique, the surgical success rate ranged between 66% and 86.6%, with the best outcomes in the majority of studies. However, it achieved similar outcomes compared with the modified ESP described by Lorusso et al. (80% vs. 90%) [14] and with ESPwAP analyzed by Babademez et al. (86.6% vs. 84.9%) [19]. The success rate of the ESP technique ranged from 64.81% to 90%. For the UPP technique, the surgical success rate was found to vary significantly among the studies, with the lowest success rate of 38.71% and the highest of 58.06%. Regarding AP, results differed considerably between studies, with Haytogiu et al. [21] reporting a success rate of 81.8%, and Karakok et al. [20] reporting a success rate of only 45%. LP obtained a success rate of 54.55% [17] and 64% [20], respectively. Being described in just one study [21] in comparison to AP, uvulopalatoflap placement obtained a favorable outcome with an overall procedural success rate of 82.6%.

### 3.4. Factors That Might Influence the Results

Having only the mean age available and not individual participant BMI limited our ability to conduct a moderation analysis using age as a separate predictor. The same was available for the BMI variable. However, we performed a mixed-effects model analysis using severity of disease as a moderator, categorizing it into two levels: mild to moderate (coded as 1) and moderate to severe (coded as 2). The intercept estimate was 1.810, while the moderator estimate was −0.349, suggesting a negative association between the severity of the disease (specifically moderate to severe) and the outcome variable (AHI mean difference). Nevertheless, this association was not statistically significant (*p* = 0.142).

### 3.5. Complications

Complications were addressed in seven studies. Table 3 provides a summary of the number of patients who experienced complications in each study, along with their corresponding citations. Lorusso et al. [14] described a persistent feeling of a foreign body in the throat, both in MESP and MBRP, that disappeared after a few months, and suggested the pain was more prolonged in the MESP group. Additionally, two cases of early bleeding were described. A lasting sensation of a foreign body was the most frequently observed complication of UPPP in the Lombo study [15] compared to BRP, which had the highest incidence of early bleeding. Tsou et al. [16] found not significantly statistically notable variations in the incidence of bleeding, dysgeusia, and globus among the two groups.

However, globus was more prevalent in the BRP group compared to the UPP group (22.58% vs. 9.67%). Martinez et al. [17] did not observe any particular complications other than minor bleeding. Postoperative pain was lower with the BRP technique compared to ESPwAP, as suggested by Babademez et al. [19] In the Karakok study [20], difficulty in swallowing and nasal regurgitation were present in all three groups, while velopharyngeal insufficiency was found only in the AP and LP groups. However, nasal regurgitation and nasal velopharyngeal insufficiency resolved within 1 month postoperatively. Postoperative bleeding was also described after the LP and ESP procedures, with two patients from the ESP group requiring reintervention. Haytogiu et al. [21] reported that postoperative pain at rest and during swallowing was significantly lower after AP compared to UPF. However, the foreign body sensation persisted for 6 months after UPF in seven patients compared to only one patient in the AP group.

## 4. Discussion

A meta-analysis published by Pang et al. in 2018 on palatal surgery for sleep apnea demonstrated a significant change in the utilization of the UPPP technique over a 17-year period, with its decreased usage in comparison to other techniques. The UPPP procedures performed during the 2011–2018 period encompassed 12.6% of all techniques used on 2715 patients compared to 25.6% during the 2001–2010 period [23]. This pattern was observable in several of the studies that we included, particularly those that gathered data retrospectively over an extended timeframe. Lombo et al. (2022) analyzed surgical techniques for sleep apnea performed in their institution between 2001 and 2020, and Martinez et al. between 2006 and 2018. In both studies, a shift away from traditional UPPP over the years and toward more focused and less invasive procedures was suggested (LP, ESP). Both authors adopted the BRP technique after 2015, as proposed by Viccini et al. [12].

The observed trend may be attributed to an improved understanding of the underlying anatomical and pathological mechanisms involved in sleep apnea, which has been facilitated by the rising utilization of drug-induced sleep endoscopy (DISE). This technique is believed to mimic natural sleep conditions more closely than alternative diagnostic methods, which may have contributed to its increasing popularity and adoption in recent years. A study by one author included in our analysis [20] also mentioned that at the beginning of his prospective study in 2011, DISE was not widely recognized or promoted at the time and, thus, he opted for alternative diagnostic procedures to visualize the obstructed site, such as the Mueller maneuver during awake nasal endoscopy. At the same time, this situation was encountered also by Lombo et al. [15] and Martinez et al. [17], who performed DISE only on some of the patients included. The majority of authors conducted a video fibroscopic examination, with or without the Mueller maneuver prior to surgery. Nevertheless, research indicates that in around 50% of patients with OSA, DISE leads to changes in surgical treatment plans when compared to awake evaluations [24]. Despite the increasing use and recognition of DISE, as well as the publication of the European position paper on DISE in 2014 and its subsequent update in 2017, there is still no wide agreement on a specific protocol for interpreting the procedure, especially regarding upper airway classification [2]. A recent review concluded that the VOTE classification was the preferred scoring system among different studies [25]. Two studies in our review used VOTE classification during DISE in assessing surgical plans and operative indications [16,19], and one study used NOHL grading [14], while the others did not mention any DISE criteria for patient selection. Nevertheless, only Lorusso et al. [14] conducted a post-surgical DISE to assess the impact of palatopharyngeal surgery on the upper airway space and confirm the appropriate selection of patients [14]. The results indicated a significant transverse reduction in oropharyngeal obstruction after both MESP and MBRP, as well as a slight decrease in hypopharyngeal obstruction, which was slightly more pronounced in the MESP group. Establishing a validated model that connects the results of DISE classification with surgical treatment plans and outcomes would provide a reliable basis for making well-informed decisions about surgical options.

Even though the UPP significantly decreased AHI in all included studies, the success rates varied highly, ranging from 38.71% to 59.26%. This can be attributed to variances in the selection criteria adopted by various studies, as well as subtle distinctions in the techniques performed. Our observations were similar to those obtained in a large systematic review published by Stock et al. that reported success/response rates between 35% and 95.2% after UPPP [26]. However, performing UPPP along with hyoid suspension significantly increases its efficacy compared to single-level surgery, through the effect it has on latero-lateral diameters of hypopharynx, widening the transverse oropharyngeal space, as Minni. et al. observed in their study [18]. The success rate rose from 40.74% to 59.26% with this combination of techniques. This finding is based on the study published in 2021 by Tessel et al. [27] that achieved noteworthy results for UPPP practiced in conjunction with HS in patients with moderate to severe OSA, with a success rate of 76.9%. Nonetheless, UPPP could have favorable outcomes as a single-level surgery in well-selected patients, as a recent meta-analysis suggests [28]. Predominantly velopharyngeal obstruction and Friedman stage 1 are predictors of success, while low hyoid position can negatively impact the surgical outcome.

The anterior palatoplasty outcomes were contradictory in the studies included in the review, as Haytogiu et al. [21] reported a success rate of 81.8% compared with only 45% obtained by Karakok et al. [20]. This was unexpected, as both authors had comparable inclusion criteria (small tonsils, retropalatal obstruction), and tonsillectomy was not performed in either of the studies. However, Karakok reported a lack of postoperative PSG for 10 patients in the AP group, which might have influenced the results. In Haytogiu’s study, the uvulopalatal flap (UPF) technique exhibited outcomes comparable to those with AP, with success rates of 82.6% and 81.8%, respectively. These findings were in line with the results obtained in another study that reported success rates of 84% and 86% for UPF and AP, respectively [29]. Notably, both studies demonstrated that patients who underwent UPF experienced higher levels of pain. This may suggest that AP could be the preferred treatment option for patients with similar characteristics.

Barbed reposition pharyngoplasty has widely grown in popularity since the first mention of the technique in 2015 by Viccini et al. [12]. A systematic review published in 2022, including all relevant articles regarding the procedure, found that barbed reposition pharyngoplasty is a simple, fast, and secure procedure for treating palatopharyngeal issues that can be used on its own or as part of a more complex surgical approach [30]. This was also reflected in our review, as BRP obtained the best results in the majority of studies in both objective and subjective parameters and mono- and multi-level procedures in patients with mild to severe disease. Moreover, as Minni et al. observed in their study, an association of hyoid suspension did not lead to an improvement in outcomes compared to UPPP (64.29% vs. 65.52%, respectively) [18]. Thus, it can be assumed that a satisfactory enlargement of the retropalatal area can be achieved without requiring the need for HS in the case of BRP, as the authors suggested. However, similar favorable results were observed in the modified version of the expansion sphincter pharyngoplasty (MESP) technique developed by Lorusso et al. [14] that demonstrated a surgical success rate of 90%, comparable with that of MRP (80%) [14,19,31]. The combination of AP with ESP in the Babademez study [19] proved to be beneficial, with a success rate of 84.9%, as it approaches both antero-posterior and latero-lateral collapse in the retropalatal space. Moreover, a randomized control trial published by Ciger et al. concluded that ESP combined with AP can effectively address various types of pharyngeal obstruction (antero-posterior, latero-lateral and concentrical) [32]. According to a 2019 multicenter study on palate surgery complications by Pang et al. [7], foreign body sensation was found to be the most prevalent complication, particularly in the UPP group. Our study corroborates this finding. Interestingly, our study revealed that many patients in the BRP group also experienced foreign body sensations. This could be attributed to the fact that most of the patients in this group with this symptom underwent a multilevel approach (BRP plus base of tongue reduction). In the LP group in our study, the most prevalent symptom was dysphagia, which aligns with Pang et al.’s statement and decision to shift toward non-invasive procedures such as ESP, initially introduced by Pang. For the BRP and ESP techniques, early bleeding was the complication most mentioned in the studies. However, this could also be the result of tonsillectomy that was performed in the same operative time. Nonetheless, a notable reduction in postoperative pain was observed with the BRP technique compared with ESP in the Lorusso and Babademez studies [14,19]. Additionally, it was noted that the operation time was shorter. A recent meta-analysis also reported that both techniques have comparable effectiveness in terms of AHI, ODI, LOS, and ESS, but BRP is faster to execute [31].

### Study Limitations

Consistent with other recently published reviews [33], we observed a lack of prospective studies directly comparing the velopharyngeal surgical techniques. Most of the studies included in this review were retrospective, which might imply an inferior level of evidence, being prone to selection bias. However, the inclusion of longitudinal studies helped us to obtain an overview of how surgical approaches changed over the years, from an aggressive approach to more reconstructive and more focused procedures. The attempt to better understand the anatomy and physiology of the upper airway in obstructive sleep apnea also determined a change in diagnostic procedures, with extended usage of DISE in the preoperative plan, which might have had an impact on surgical results in some studies. The longitudinal nature of the studies and the accumulation of experience in performing DISE over time allowed for the refinement of patient selection for surgery, which may have had an impact on the surgical outcomes.

## 5. Conclusions

A trend toward adopting the BRP surgical technique has emerged due to its excellent outcomes with minimal complications and a relatively short operation time. Our results show that BRP achieved the most favorable objective and subjective outcomes among the studies reviewed, with consistent results in both single and multilevel settings. Although ESP closely follows BRP in terms of efficiency, it comes with a higher risk of complications such as postoperative prolonged pain and shows better results when combined with techniques such as AP that target antero-posterior collapse. However, other, older techniques, such as UPPP, UPF, AP, or LP in mono- or multilevel settings, also have good results when performed in well-selected patients. The heterogeneity observed in the studies, both in terms of surgical outcomes and patient selection criteria, led us to conclude that more prospective studies with standardized selection criteria for each group of patients are needed to generalize the outcomes. Despite the well-established role of DISE worldwide, there is still a lack of specific consensual criteria regarding the upper airway classification of obstruction patterns. The introduction of such criteria may assist surgeons in selecting the appropriate therapeutic strategy for each patient and researchers in obtaining more accurate and generalizable results.

## Figures and Tables

**Figure 1 medicina-59-01147-f001:**
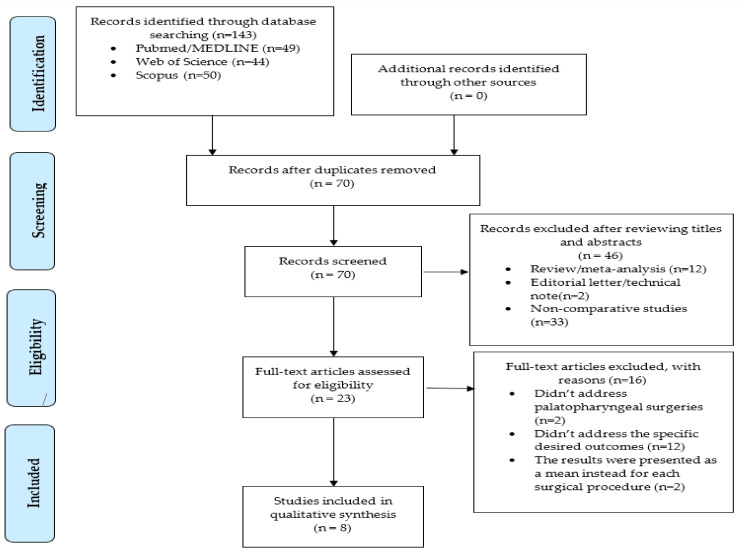
PRISMA flow diagram.

**Figure 2 medicina-59-01147-f002:**
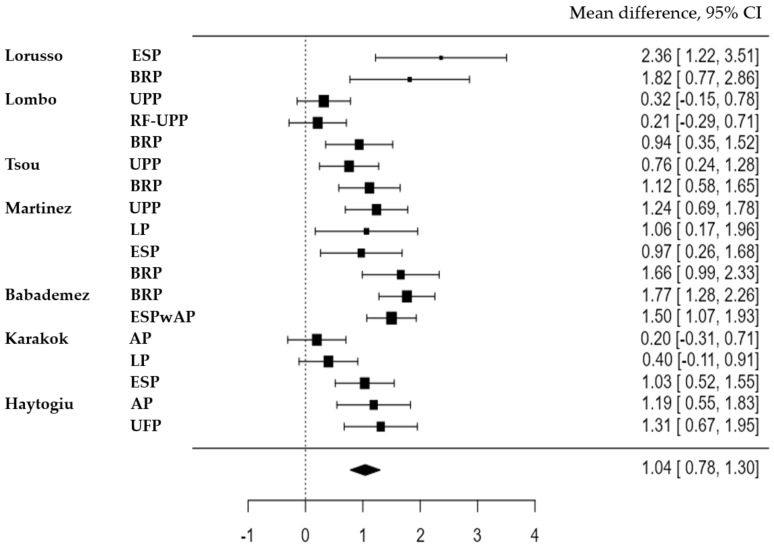
Forest plot AHI-Comprison between pre- and postoperative results.

**Figure 3 medicina-59-01147-f003:**
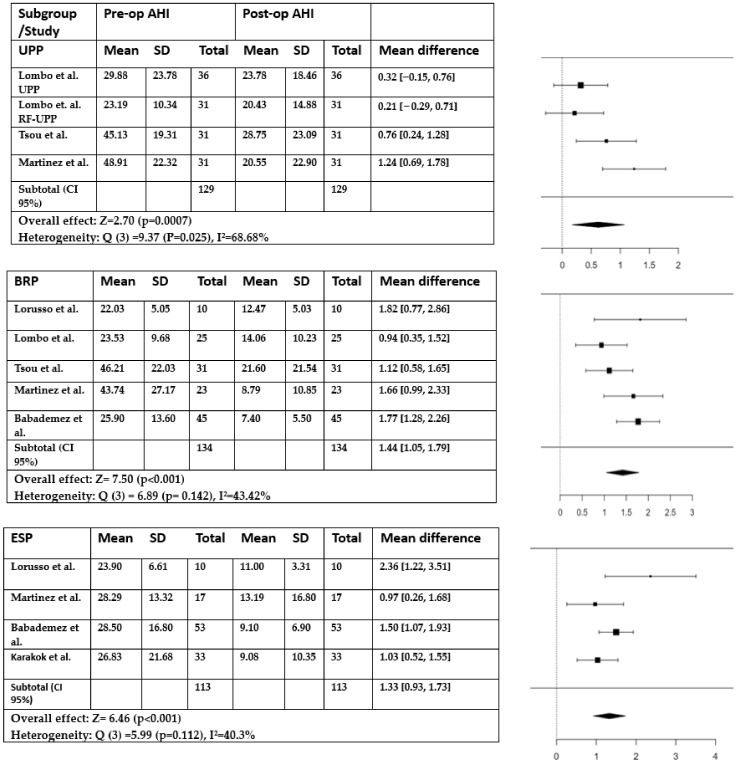
Subgroup analyses of AHI outcomes using random-effects model.

**Figure 4 medicina-59-01147-f004:**
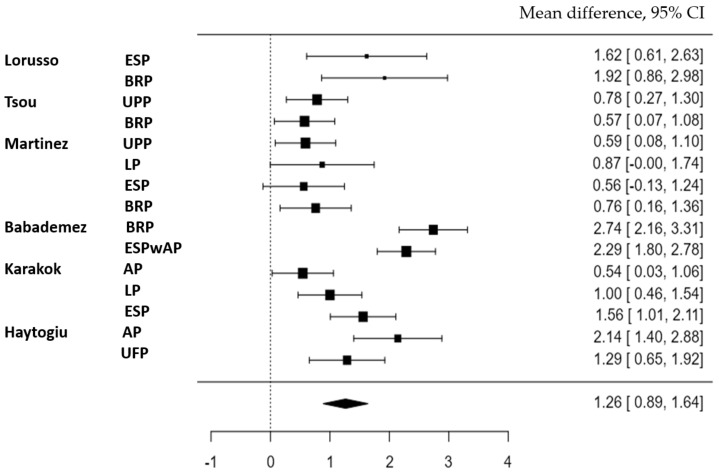
Forest plot ESS—Comparison between pre- and postoperative results.

**Table 1 medicina-59-01147-t001:** Main characteristics of the included studies.

Author (Year)	Study Design	Patient Number	Mono-/ Multilevel	Follow-up	Mean Age	Sex (M:F)	BMI	Snoring/OSA	DISE
Lorusso et al., 2022 [14] Italy	Retrospective study	MESP = 10 MBRP = 10	Monolevel	12 months	48.4 ± 4.8 41.6 ± 11.77	20M:0F	28.4 ± 3.06 28.7 ± 3.02	Mild to moderate OSA	Yes
Lombo et al., 2022 [15] Portugal	Retrospective study	UPP=36 RF-UPP=31 BRP=25	Monolevel	12 months	49.36 ± 9.6	85M:7F	29.14 ± 2.94	OSA + snorers	Yes
Tsou et al., 2021 [16] China	Retrospective study	UPPP = 31 BRP = 31	Multilevel + TORSBTR	6 months	39.61 ± 11.63 37.51 ± 9.42	24M:7F 26M:5F	28.20 ± 3.62 28.22 ± 3.19	Moderate to severe OSA	Yes
Martinez et al., 2020 [17] Spain	Retrospective study	UPP = 31 LP = 11 ESP = 17 BRP = 23	Monolevel	4 months	42.0 ± 19.78	70M:12F	27.63 ± 3.7	Moderate to severe OSA	Yes
A. Minni et al., 2021 [18] Italy	Retrospective study	UPPP = 80 BRP = 42	Mono/multilevel (±HS)	18 months	43 (37–47) 42 (38–47)	51M:29F 20M:22F	25 (24–26) 27 (25–28)	Moderate to severe OSA	No
Babademez et al., 2020 [19] Turkey	Retrospective study	BRP= 45 ESPwAP = 53	Monolevel	18.8 months (median)	37.3 ± 8.9 41.6 ± 9.4	31M:14F 41M:12F	29.3 ± 3.1 28.8 ± 4.2	Mild to severe OSA	Yes
Karakok et al., 2018 [20] Turkey	Prospective study	AP = 30 LP = 30 ESP = 33	Monolevel	5.90 ± 6.23 months	40.7 ± 9.59	27M:3F 30M:0F 32M:1F	27.67 ± 2.96	OSA + snorers	No
Haytogiu et al., 2018 [21] Turkey	Prospective study	AP = 22 UFP = 23	Monolevel	6 months	39.2 41.3	12M:10F 14M:9F	28.0 ± 1.6 27.3 ± 1.8	Mild to moderate OSA	No

UPP—classical uvulopalatoplasty, UPPP—modified uvulopalatoplasty, LP—lateral pharyngoplasty, ESP—expansion sphincter pharyngoplasty, MESP—modified ESP, BRP—barbed reposition pharyngoplasty, MBRP—modified BRP, AP—anterior palatoplasty, UFP—uvulopalatal flap placement, HS—hyoid suspension, TORSBTR—transoral robotic base of tongue reduction, BMI—body mass index, DISE—Drug Induced Sleep Endoscopy.

**Table 2 medicina-59-01147-t002:** Included studies outcomes.

Author (Year)	Surgical Techniques	Pre-op AHI	Post-op AHI	Pre-op ESS	Post-op ESS	Success Criteria	Success Rate
Lorusso et al., 2022 [14] Italy	**MESP** vs. **MBRP**	23.9 ± 6.61 vs. 22.03 ± 5.05	11 ± 3.3 **a** vs. 12.47 ± 5.03 **a**	10.4 ± 3.1 vs. 9.1 ± 2.07	5.1 ± 3.17 **a** vs. 4.5 ± 2.5 **a**	Sher criteria AHI reduction > 50 and AHI value < 20	90% vs. 80%
Lombo et al., 2022 [15] Portugal	**UPP (classical)** vs. **RF-UPP** vs. **BRP**	29.88 ± 19.40 vs. 23.19 ± 10.34 vs. 23.53 ± 9.68	23.78 ± 18.46 **a** vs. 20.43 ± 14.88 **a** vs. 14.06 ± 10.23 **a**	nd	nd	Sher criteria AHI reduction > 50 and AHI value < 20	57% vs. 54% vs. 66%
Tsou et al., 2021 [16] Chins	**UPPP (modified)** vs. **BRP**	45.13 ± 19.31 vs. 46.21 ± 22.03	28.75 ± 23.09 **a** vs. 21.60 ± 21.54 **a**	11.01 ± 4.52 vs. 9.03 ± 4.52	7.82 ± 3.45 **a** vs. 6.60 ± 3.82 **a**	Sher criteria AHI reduction > 50 and AHI value < 20	38.71% vs. 67.74%
Martinez et al., 2020 [17] Spain	**UPP (classical)** vs. **LP** vs. **ESP** vs. **BRP**	48.91 ± 22.32 vs. 46.3 ± 34.02 vs. 28.29 ± 13.32 vs. 43.74 ± 27.17	20.55 ± 22.9 **a** vs. 17.84 ± 13 **a** vs. 13.19 ± 16.8 **a** vs. 8.79 ± 10.85 **a**	9.6 ± 4.95 vs. 9.78 ± 4.21 vs. 7.12 ± 5.43 vs. 8.33 ± 4.7	6.89 ± 4.1 **a** vs. 6.4 ± 3.2 vs. 4.54 ± 3.33 vs. 5.19 ± 3.3 **a**	Sher criteria AHI reduction > 50 and AHI value < 20	58.06% vs. 54.55% vs. 64.71% vs. 78.26%
Minni et al., 2021 [18] Italy	**UPPP (modified)** vs. **UPPP + HS** vs. **BRP** vs. **BRP + HS**	27 (24–29) vs. 27 (24–29) vs. 29 (28–31) vs. 28 (26–30)	16 (14–17) **a** vs. 11 (10–11)**a** vs. 10 (9–11) **a** vs. 10 (9–11) **a**	12 (12–13) vs. 13 (12–13) vs. 13 (12–13) vs. 13 (12–13)	12 (11–12) vs. 11 (11–12) vs. 10 (10–11) vs. 11 (10–12)	AHI < 20, ESS < 10, both reduced > 50%	nd
Babademez et al., 2020 [19] Turkey	**BP** vs. **ESPwAP**	25.9 ± 13.6 vs. 28.5 ± 16.8	7.4 ± 5.5 **a** vs. 9.1 ± 6.9 **a**	11.2 ± 3.7 vs. 12.6 ± 4.9	3.4 ± 1.5 **a** vs. 4.1 ± 1.8 **a**	Sher criteria AHI reduction > 50 and AHI value < 20	86.6% vs. 84.9%
Karakok et al., 2018 [20] Turkey	**AP** vs. **LP** vs. **ESP**	16.90 ± 10.26 vs. 17.69 ± 12.47 vs. 26.83 ± 21.68	14.27 ± 15.43 vs. 12.05 ± 15.23 **a** vs. 9.08 ± 10.35 **a**	9.35 ± 4.67 vs. 13.21 ± 4.89 vs. 11.06 ± 5.21	6.80 ± 4.59 vs. 8.28 ± 4.84 **a** vs. 4.25 ± 3.19 **a**	Modified Sher criteria AHI reduction > 50 and AHI value < 15	45% vs. 64% vs. 74%
Haytogiu et al., 2018 Turkey [21]	**AP** **vs.** **UFP**	17.5 ± 8.2 vs. 18.5 ± 7.9	8.1 ± 7.3 **a** vs. 8.6 ± 6.9 **a**	13.6 ± 3.3 vs. 10.8 ± 3.3	6.4 ± 3.3 **a** vs. 5.4 ± 4.8 **a**	Sher criteria AHI reduction > 50 and AHI value < 20	81.8% vs. 82.6%

UPP—classical uvulopalatoplasty, UPPP—modified uvulopalatoplasty, LP—lateral pharyngoplasty, ESP—expansion sphincter pharyngoplasty, MESP- modified ESP, BRP—barbed reposition pharyngoplasty, MBRP—modified BRP, AP—anterior palatoplasty, UFP—uvulopalatal flap placement, HS—hyoid suspension, **a**—of statistical significance as reported by authors, nd—not determined, AHI—apnea–hypopnea index, ESS—Epworth sleep scale.

**Table 3 medicina-59-01147-t003:** Postoperative complications for different techniques.

	Techniques						
Complications	UPPP	RF-UPP	BRP	ESP	LP	AP	UFP
Foreign Body Sensation/Globus	8 [15] 3 [16]	5 [15]	2 [14] 2 [15] 7 [16]	2 [14]	-	1 [21]	8 [21]
Nose regurgitation	-	-	-	-	2 [20]	2 [20] 1 [21]	-
Velopharyngeal insufficiency	-	-	-	-	1 [20]	1 [20]	-
Prolonged pain	3 [15]	1 [15]	1 [15]	-	-	-	-
Early bleeding	2 [15] 2 [16]	-	4 [15] 1 [16]	2 [14] 4 [20]	2 [20]	-	-
Suture dehiscence	1 [15]	-	2 [15]	-	-	-	-
Dysphagia	6 [16]	-	5 [16]	7 [20]	9 [20]	3 [20]	-
Dysgeusia	1 [16]	-	1 [16]	-	-	-	-

## Data Availability

Not applicable.

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
