# Peer review of "Comparative Efficacy of Velopharyngeal Surgery Techniques for Obstructive Sleep Apnea: A Systematic Review"

_medicina, 2023, doi:10.3390/medicina59061147_

Round 1

Reviewer 1 Report

In my opinion, the work needs major improvements. Here below are listed my comments:

1.       The fonts in the main body need to be uniform.

2.       The figure need to be fully displayed.

3.       Age may have an impact on surgical outcomes. For the analysis of the findings,the age of all patients need to be considered,not just the average age.

4.       Postoperative recovery and complications may depending on the severity of the disease. I hope you could take these factors into consideration.

5.       Different surgical recommendations should be made for various causes or severity of sleep apnea disease based on research.

 The paper needs an extensive review for spelling, grammar, and readability.

Reviewer 2 Report

This manuscript is a review on the Efficacy of Velopharyngeal Surgery Techniques for Obstructive Sleep Apnea: A Systematic Review.

I recommend its publication after major revision.

The authors have selected 8 papers comparing several velopharyngeal techniques, although they were heterogenous, not including postop PSG findings, and combining retrospective to prospective studies. However, the results the present clearly favor BRP, as current literature also show (see ref 32).

The text needs a full review, to make it clearer and more precise. Tables should be improved. Figures comparing results are needed. A meta-analysis could be done...

Round 2

Reviewer 2 Report

All suggestions have been amended, including the addition of a meta-analysis. 

Please, review the whole text to remove coloquial terms.